# Promising Gastric Cancer Biomarkers—Focus on Tryptophan Metabolism via the Kynurenine Pathway

**DOI:** 10.3390/ijms26083706

**Published:** 2025-04-14

**Authors:** Kinga Ożga, Paweł Stepuch, Ryszard Maciejewski, Ilona Sadok

**Affiliations:** 1Department of Biomedicine and Environmental Research, Institute of Biological Sciences, Faculty of Medicine, Collegium Medicum, The John Paul II Catholic University of Lublin, Konstantynów 1J, 20-708 Lublin, Poland; kinga.ozga@kul.pl; 2II Department of Oncological Surgery with Subdivision of Minimal Invasive Surgery, Center of Oncology of the Lublin Region St. Jana z Dukli, Jaczewskiego 7, 20-090 Lublin, Poland; stepuch.p@gmail.com; 3Faculty of Medicine, Collegium Medicum, The John Paul II Catholic University of Lublin, Konstantynów 1H, 20-708 Lublin, Poland; ryszard.maciejewski@kul.pl; 4Department of Biomedical and Analytical Chemistry, Institute of Biological Sciences, Faculty of Medicine, Collegium Medicum, The John Paul II Catholic University of Lublin, Konstantynów 1J, 20-708 Lublin, Poland

**Keywords:** metabolomic analysis, proteomic analysis, gastrointestinal cancers, kynurenine, inflammation

## Abstract

Currently, gastric cancer treatment remains an enormous challenge and requires a multidisciplinary approach. Globally, the incidence and prevalence of gastric cancer vary, with the highest rates found in East Asia, Central Europe, and Eastern Europe. Early diagnosis is critical for successful surgical removal of gastric cancer, but the disease often develops asymptomatically. Therefore, many cases are diagnosed at an advanced stage, resulting in poor survival. Metastatic gastric cancer also has a poor prognosis. Therefore, it is urgent to identify reliable molecular disease markers and develop an effective medical treatment for advanced stages of the disease. This review summarizes potential prognostic or predictive markers of gastric cancer. Furthermore, the role of tryptophan metabolites from the kynurenine pathway as prognostic, predictive, and diagnostic factors of gastric cancer is discussed, as this metabolic pathway is associated with tumor immune resistance.

## 1. Introduction

Gastrointestinal cancer is a disease that is varied in biological, morphological, and genetic terms. GLOBOCAN 2020 reports that gastric cancer (GC) is the fifth most common malignant tumor worldwide and the fourth leading cause of cancer death among all cancers. The predictions for future incidence of and mortality from GC are not optimistic (Figure 1). Mortality from GC is high due to delayed diagnosis (often in the advanced stage) [1]. During 2017–2021, the estimated 5-year relative survival rates for male and female patients with GC were 38.3% and 50.4%, respectively [2]. The incidence of GC is almost two times higher in men than in women and is correlated with increasing age. Despite this, young people under 50 are increasingly being diagnosed with GC [1]. The general categorization of GC includes two subtypes: cardia (common in Western countries) and noncardia (common in Eastern Asia) [3]. Most cases of GC are adenocarcinomas, while lymphoma, sarcomas, and carcinoids are less frequent (<5% of all cases) [4].

Only early diagnosis improves the prognosis of patients with GC. Symptoms of GC include dyspepsia, epigastric pain, nausea, emesis, weight loss, anorexia, and early satiety. These are nonspecific signs that frequently manifest in the advanced stages of GC. Therefore, many patients are diagnosed too late and have distant or local metastases. Currently, GC diagnosis uses endoscopy with a thorough evaluation of the gastric mucosa, biopsy of suspicious lesions, and computerized tomography. Although endoscopy is still the gold standard for the diagnosis of GC, patients tend to avoid it because it is uncomfortable. Moreover, this invasive examination has possible side effects including perforation or hemorrhage [6]. Therefore, the development of an improved detection method for GC diagnosis in the early stages is in demand. In regions with a higher incidence of GC, such as Japan and the Far East, there are screening programs in which patients over a certain age undergo gastroscopy. Such initiatives result in a higher percentage of lesions being detected in the early phase. This type of preventive program does not exist in Europe. It is of great clinical importance to find a biomarker that would encourage deeper diagnosis and implementation of gastroscopy in justified cases (such as prostate-specific antigen (PSA) in prostate disease) and would provide the possibility of monitoring the course of the disease.

A biomarker is a biological observation that allows easy and inexpensive prediction of a clinically relevant endpoint or intermediate outcome. In the case of cancer treatment, biomarkers are used, among other reasons, to assess the risk of cancer, evaluate tumor–host interactions, determine tumor burden, and analyze cell function, as well as to predict tumor responses to medications [7]. Currently, only a few biomarkers are clinically used for early detection of GC, such as carbohydrate antigens (CA19-9, CA72-4, CA125, CA24-2, CA50), carcinoembryonic antigen (CEA), pepsinogen, and α-fetoprotein (AFP). However, none of them is effective enough due to their poor specificity and sensitivity [8]. Furthermore, an individual biomarker cannot provide all the information necessary to monitor interventions [7]. Therefore, further advances in the field of molecular biology could lead to the identification of new sophisticated biomarkers and improved treatment options.

Immunotherapy is likely to be the future of GC treatment [9]. Inflammatory tissue microenvironments contribute strongly to tumor progression [10]. A pioneering study on maternal immune tolerance by Munn et al. [11] yielded novel insights into the role of tryptophan (TRP) catabolism via the kynurenine pathway (KP) in tumor resistance to immune rejection. The activity of three heme-containing enzymes–tryptophan 2,3-dioxygenase (TDO) and indoleamine 2,3-dioxygenase 1 and 2 (IDO1 and IDO2, respectively)– can suppress antitumor immunity via local reduction of TRP access and secretion of kynurenine (KYN) and other KP metabolites [11].

The KP is responsible for >95% of TRP degradation (Figure 2) and leads to de novo synthesis of NAD+ and NADP+ [12]. The first step in the KP is cleavage of the indole ring of TRP and its conversion to N′-formykynurenine (NFK), mediated by IDO1, IDO2, and TDO. In particular, IDO1 and TDO are essential factors in immune regulation [13]. TDO (formerly tryptophan pyrrolase) is normally expressed almost exclusively in the liver and is responsible for ~90% of the total degradation of TRP under normal physiological conditions [14]. The extrahepatic KP, regulated by IDO isoforms, contributes only <2% to this process under normal conditions, but is significantly enhanced after immune activation. N′-formylkynurenine is hydrolyzed to kynurenine (KYN), identified as a specific agonist of the human aryl hydrocarbon receptor (AHR), activation of which promotes invasiveness and accelerates growth of cancer cells [15]. Further, in the KP, KYN is mainly hydrolyzed by kynurenine monooxygenase (KMO) to 3-hydroksykynurenine (3HKYN), which can be transformed into xanthuerenic acid (XA) by kynurenine aminotransferase (KAT). Alternately, KYN can be metabolized to kynurenic acid (KYNA) by kynurenine aminotransferase or anthranilic acid (AA) by kynureninase [16]. Another downstream KP metabolite, 3-hydroxyanthranilic acid (3HAA), can be produced from AA through a non-enzymatic reaction or from 3HKYN by the activity of kynureninase B. In the next step, 3HAA is transformed into 2-amino-3-carboxymuconic acid-6 semialdehyde (ACMS) by 3-hydroxyanthranilic acid-3,4-dioxygenase (3-HAAO) or oxidized to cinnabarinic acid (CA). At this stage, the KP branches into two arms. ACMS is rapidly converted to quinolinic acid (QA) by the non-enzymatic reaction or to 2-aminomuconic semialdehyde (AMS) [17]. In the KP, QA is further processed by QA phosphoribosyl transferase (QPRT) to the precursor nicotinic acid mononucleotide (NAMN) and then to NAD+ [16]. AMS is transformed to picolinic acid (PIC) by non-enzymatic cyclization or 2-aminomuconic acid by 2-aminomuconate semialdehyde dehydrogenase, which can be further degraded to acetyl-CoA [18].

Inhibition of key KP enzymes promotes tumor immune rejection, an attractive feature for cancer immunotherapy [19]. Furthermore, TRP metabolites can promote tumor progression by suppressing antitumor immune responses and increasing the malignant properties of cancer cells [20]. So far, several studies have evaluated the utility of KP metabolites as potential biomarkers in GC [21,22,23,24]. This review summarizes clinically used GC biomarkers, as well as putative prognostic and predictive markers. Special emphasis is placed on KP metabolites and their potential role in GC pathogenesis, diagnosis, and treatment.

## 2. Gastric Cancer Etiology and Diagnosis

Despite GC having multifactorial etiologies, *Helicobacter pylori* infection and gene polymorphisms, diet, and lifestyle habits are the main predisposing factors [6,18]. Highly processed foods including salt-preserved foods, as well as smoked foods, grilled (broiled) and barbecued (charbroiled) animal meats, and foods high in N-nitroso compounds may contribute to the development of cancer. On the other hand, a diet rich in fruits and vegetables might protect against GC tumorigenesis [12,15]. Infection with Epstein–Barr virus is responsible for about 10% of GC cases. This viral infection can promote carcinogenesis via inflammation of the gastric mucosa or hypermethylation of tumor suppressor genes [3]. However, *H. pylori* infection is one of the main risk factors for GC. Chronic infection with *H. pylori* alters gastric acid secretion, causing loss of acidity that exacerbates inflammation and also leads to gastric ulcers, atrophy, metaplasia, and dysplasia. GC risk correlates with the duration of bacterial infection [3,25,26].

About 10% of all GC cases aggregate in families, but only 1–3% are hereditary. The risk of GC is increased in people who have mutations in genes such as p5><YUHiop3, BRCA2, MSH2, MLH1, CDH1, CTNNA1, PALB2, FBXO24, PSCA, MUC1, and PLCE1 [27,28,29]. Furthermore, this type of cancer often occurs in individuals with genetic diseases such as Peutz–Jeghers syndrome, familial adenomatous polyposis, Lynch syndrome, Cowden syndrome, juvenile polyposis, Li–Fraumeni syndrome, and MUTYH-associated adenomatous polyposis [20,21]. Radiation exposure, older age, male sex, lack of physical activity, and obesity are also risk factors for GC [12,18].

There are often no specific symptoms of GC in the advanced stage. Therefore, the majority patients are diagnosed too late and have distant or regional metastases. In particular, complete recovery is possible only in the case of early diagnosis of the disease.

GC is a heterogeneous disease histologically, molecularly, and immunologically. There are four types of GC: intestinal, diffuse, mixed, and unclassified, while the WHO has specified more subclasses of stomach adenocarcinomas [30]. Additionally, the Cancer Genome Atlas Research Network had proposed four molecular GC subtypes: Epstein–Barr virus–associated (EBV), microsatellite instable (MSI), chromosomal instable (CIN), and genomically stable (GS). The 5-year survival rates are only 60–80% and 5% for the I and IV stages of the disease, respectively [3]. Pathogenesis based on atrophic gastritis, then intestinal metaplasia, followed by dysplasia and neoplastic transformation, leads to the development of the so-called intestinal type of GC, according to Lauren’s classification. This type of tumor is characterized by a better prognosis and is typically located distally, making it more feasible to perform less extensive resection surgeries. In recent years, a decline in the incidence of this histological type of cancer has been observed, which is often attributed to the effective treatment of gastric mucosal inflammation, notably due to the introduction of proton pump inhibitors.

Diffuse GC spreads intramurally and is associated with a poor prognosis. For this type of GC, many oncological centers tend to opt for gastroscopy, i.e., total stomach resection, as the preferred treatment procedure to ensure complete tumor resection. The Polish consensus on the diagnosis and treatment of GC (update from 2022) allows a minimum resection margin of 5 cm regardless of the histological type of cancer [31,32].

Currently, GC diagnosis uses endoscopy with thorough evaluation of the gastric mucosa, biopsy of suspicious lesions, and computerized tomography. However, laparoscopy followed by the histopathological classification by Lauren is used for peritoneal staging of GC [33]. Endoscopy is still the gold standard for the diagnosis of GC [6]. Thanks to the latest developments in endoscopic technology, it is possible to accurately assess the gastric mucosa. Various image-enhanced endoscopy (IEE) techniques have been developed to complement conventional gastroscopy, including blue laser imaging (BLI), linked color imaging (LCI), narrow-band imaging (NBI), and texture and color enhancement imaging (TXI). NBI increases the effectiveness of endoscopy, especially in detecting lesions in the early phase of the disease. NBI imaging is also very useful in diagnosing gastroesophageal junction cancers, allowing determination at the endoscopy stage of the type of epithelium from which the cancer originates (glandular or stratified squamous epithelium). A recent meta-analysis confirmed an improved GC detection rate using IEE (especially LCI) compared to white-light imaging [34]. Furthermore, endoscopy is increasingly used to determine the presence/absence of *H. pylori* infection. The Kyoto classification categorizes endoscopic observations associated with *H. pylori* infection. A score of 4 or more accompanied by atrophy, intestinal metaplasia, enlarged folds, and nodularity may suggest an elevated GC risk [35]. Despite advances in endoscopic techniques, they are still considered invasive, require anesthesia, and carry a small but significant risk of complications, including perforation or hemorrhage [6,36]. Furthermore, endoscopic dissections are reserved only for early-stage GC (T1a), where the tumor infiltrates the lamina propria or the muscularis mucosae without invading the submucosa, and the diameter of the lesion should not exceed 2 cm. Unfortunately, in countries without screening programs, GC is very rarely diagnosed at this stage.

Magnetically controlled capsule gastroscopy (MCCG) can be a good alternative because it is non-invasive and painless. It involves the patient swallowing a capsule equipped with a camera that takes a series of photographs, which are then analyzed by the doctor. Diagnostic sensitivity and accuracy rates of MCCG sometimes exceed 90%, showing great effectiveness in the screening of GC and precancerous conditions [36]. Capsule endoscopy is a modern examination usually used when conventional endoscopic methods are not feasible. Currently, there are also reports of successful image analysis by artificial intelligence. However, its use in the diagnosis of GC is significantly limited, primarily due to its reliance on a non-inflated stomach, the inability to assess susceptibility to insufflation, and the lack of assessment of the mucous membrane throughout the entire stomach. Furthermore, smaller lesions located between the folds may not be visible. This technique also does not allow for the collection of biopsies or histopathological examination.

In the case of more advanced cancers, various resection surgeries are performed, ranging from organ-sparing operations such as subtotal gastrectomy to gastrectomy, which is complete removal of the stomach. Regardless of the extent of gastric resection, due to lymphatic spread of the tumor and early metastasis to local lymph nodes, the foundation of surgery is a properly performed operation on the lymphatic system. The extent of lymphadenectomy is strictly defined and tailored to the stage of the disease.

Surgery is a highly effective approach to treatment, particularly in the early stage of GC. In advanced cases of GC (>cT1b) and in patients who qualify for radical surgical treatment, standard D2 lymphadenectomy is recommended. D1/D1+ lymphadenectomy is allowed in patients who qualify for surgical treatment for early-stage cancer (cT1a), as well as in the case of palliative resections. Chemotherapy and chemoradiotherapy are useful for supporting the effects of surgery [3,30]. Importantly, GC is characterized by resistance to chemotherapy and drugs. Chemotherapy is used most frequently in patients with advanced metastases, but the response to chemotherapy could be poor. The exact mechanism is not yet clear. It is probably caused by accelerated cell proliferation and autophagy flux, enhanced DNA damage repair capacity of cancer cells or inactivation of apoptosis signaling pathways, and loss of cell cycle checkpoint control [37]. Therefore, it is important to identify new biomarkers for the early diagnosis of GC [3].

## 3. Established Biomarkers of Gastric Cancer

In clinical practice, only a few biomarkers are used for the early detection of GC, but their specificity and sensitivity are insufficient, thus requiring several markers to be tested simultaneously [38]. Carcinoembryonic antigen (CEA) is a nonspecific serum biomarker, since it appears at elevated levels in different malignant cancers, including colorectal cancer, medullary thyroid cancer, breast cancer, mucinous ovarian cancer, pancreatic cancer, lung cancer, and advanced stages of GC. Therefore, the monitoring of serum CEA is not effective for routine screening [39]. Other serum tumor markers, carbohydrate antigens (e.g., CA19-9, CA72-4, CA125), are widely applied in diagnosis, evaluation of treatment effects, and disease monitoring. CA19-9 is the most popular marker used in gastroenteric cancer, and its elevated level is associated with advanced stage, differentiated histology, and a higher percentage of lymph node metastases [40]. Patients with GC have high serum levels of CA72-4, which increase as the disease progresses. Treatment effectiveness and GC relapses can be monitored with this marker. However, the role of CA72-4 as a screening marker for GC remains unclear [41]. The prognostic and diagnostic significance of AFP and CA125 is of little importance, since their levels are relatively low in early GC [38]. However, they become more important in patients with GC who have metastases and a poor prognosis.

Currently, human epidermal growth factor receptor 2 (HER2) and PD-L1 (ligand of Programmed Death-1) are new predictive biomarkers, but they have some disadvantages and limitations. HER2 is the first official predictive biomarker in personalized treatment of GC. Its overexpression/amplification is correlated with advanced-stage GC [9,30]. In HER2-positive GC, treatment with the monoclonal antibody trastuzumab in combination with platinum-fluoropyrimidine first-line chemotherapy increases cure rates [42]. PD-L1 is a member of the B7 family of immune checkpoint molecules expressed by lymphocytes and dendritic cells [30]. However, PD-L1 and HER2 are characterized by high intratumoral heterogeneity. Therefore, to obtain an accurate diagnosis, it is necessary to analyze multiple samples collected from the same patient. At the moment there is a huge need to develop standardized and objective methods for PD-L1 evaluation [43,44].

## 4. Putative Biomarkers of Gastric Cancer

Biomarkers can be identified at molecular, cellular, tissue, organ, or whole-body level [7]. Exhaustive research in the fields of genomics, proteomics, and metabolomics has resulted in the identification of numerous candidates for molecular GC biomarkers, such as GC-related oncogenes, overexpressed or downregulated genes, proteins, and immune checkpoint molecules [45]. These findings might bridge the gap between research and clinical practice. For example, proteomics-based techniques are useful for identifying GC-related proteins or for searching for drug-sensitive and non-sensitive predictive biomarkers [37,38,39,40]. Most of these molecular markers are difficult to identify because of the lack of simple and validated diagnostic methods. Some of the important putative markers for gastric cancer are listed in Table 1 and summarized in Figure 3.

Some long noncoding RNAs >200 nucleotides in length (lncRNAs), which exist in body fluids, have been described as candidate biomarkers. lncRNAs are involved in many diseases, including GC. During carcinogenesis, lncRNAs may regulate GC apoptosis, proliferation, migration, metastasis, and angiogenesis. The clinical application of lncRNAs is limited by their heterogeneity and the need for rapid isolation and detection [53].

Currently, trials are being conducted to test potential inhibitors of some receptor tyrosine kinases such as HER2, hepatocyte growth factor receptor (MET), fibroblast growth factor receptor (FGFR2), and epidermal growth factor receptor (EGFR) that are overexpressed in CIN GC (Table 1) [9,30]. However, MET and HER2 show intratumoral heterogeneity that limits their applicability as predictive biomarkers [30].

Immune evasion is a strategy for malignant tumors to escape destruction by the immune system [61]. Immune checkpoint molecules gained considerable attention as novel treatment options in GC due to their involvement in immune evasion. Among this group, V-domain immunoglobulin-containing suppressor of T cell activation (VISTA) and B3-H7 are also noteworthy as GC biomarkers. VISTA is expressed by myeloid, granulocytic, and T cells, while B3-H7 may inhibit CD8+ T cells. However, none of them has reached clinical application, and sexual dimorphism in immune response capacity should be considered [30].

Mucins (high molecular weight glycoproteins expressed throughout the gastrointestinal tract) have also been evaluated for their prognostic value in GC [58]. High expression of mucin proteins such as MUC1 is associated with significantly poorer survival and aggressive pathologic features of GC, including diffuse-type cancer, advanced cancer, lymph node metastases, and distant metastases. However, the prognostic significance of MUC1 in patients with GC remains to be elucidated [57].

## 5. The Role of Kynurenine Pathway Enzymes in Tumorigenesis

KP enzymes have been an interesting subject of research, as they determine the levels of TRP metabolites through their activity and expression of their genes. Although the KP exists primarily in the liver, its downstream metabolites are distributed to different tissues and organs. Differences in the expression of KP enzymes are widely observed in normal and diseased tissues (Table 2). Notably, up-regulation or down-regulation of the KP enzyme may cause disorders that could begin tumorigenesis [16,62,63]. For example, Perez-Castro et al. analyzed KP enzyme mRNA and suggested that there is a link between the expression of TRP-metabolizing enzymes and various types of cancer. In this way, KP enzymes may play a prognostic role in a variety of cancers [63].

TDO, IDO1, and IDO2, which catalyze the first and rate-limiting step of TRP degradation via the KP, leading to NFK generation (Figure 2), are involved in tumor development and immune modulation. TDO is a homotetrameric heme-containing cytosolic enzyme encoded by a gene that is expressed at high levels in the liver. TDO is structurally distinct from IDO1 and IDO2 and has a unique function in cancer [19,69]. Activation of both TDO and IDO is suggested to promote cancer cell migration. Expression of TDO is observed in a variety of cancers (Table 2), and, unlike IDO, it is strongly activated in brain cancers. In gliomas, TDO up-regulation promotes growth of the malignancy and correlates with the proliferation index of tumor cells. Furthermore, constitutive expression of TDO in glioma cells produces KYN in amounts sufficient for AHR activation. The TDO–KYN–AHR signaling pathway is also activated in colorectal ovarian carcinoma, bladder carcinoma, lung carcinoma, cervix carcinoma, Ewing sarcoma, and B-cell lymphoma [15]. Thus, pharmacological inhibition of TDO represents a novel therapeutic approach in cancer therapy, since it could support the antitumor immune response. Therefore, the development of new inhibitors of TDO is an active research area [87,88].

IDO, which controls the extrahepatic branch of the KP, has two isoforms––IDO1 and IDO2––that differ in some biological functions. IDO1 is a monomeric heme-containing cytosolic enzyme that plays a significant role in tumor progression [19]. IDO1 is strongly implicated in tumor immune tolerance and immune escape [69]. Its role involves the suppression of natural killer (NK) cells and CD8+ T effector cells and increasing the activity of CD4+ Treg cells and myeloid-derived suppressor cells (MDSC) [89]. In addition, IDO1 contributes to cancer development by promoting neovascularization and production of the pro-inflammatory cytokine IL-6, which leads to tumor immune escape [10]. Overexpression of this enzyme is observed in 61% of human tumors (examples are given in Table 2) and is associated with a poor prognosis. For example, IDO1 overexpression occurs in GC cells and is correlated with deeper invasion and a higher frequency of lymph node metastasis [90]. The expression of IDO is also associated with the expression of the transcription factor Fork head box P3 (FoxP3) in both cell lines and clinical samples. The key role of FoxP3 transcription is the induction of immunosuppressive regulatory T cell function [91]. Blocking IDO1 activity can reduce tumor proliferation and enhance the efficacy of radiotherapy, immunotherapy, and chemotherapy. Thus, IDO1 inhibition is in demand. Although many different IDO1 inhibitors have been developed, including both competitive TRP derivatives and non-competitive compounds, the most popular is 1-methyl-D-tryptophan containing the indole ring of TRP [92]. Catalytic inhibitors of IDO1 can defeat immune escape and broadly enhance other therapeutic modalities; therefore, they have entered clinical trials [89]. However, a promising phase three clinical trial, ECHO-301/KEYNOTE-252, which evaluated epacadostat as an IDO1 inhibitor in combination with pembrolizumab, ultimately failed. The other two clinical trials also in phase three (NCT03386838 and NCT03417037) that evaluated the IDO1 inhibitor BMS-986205 in combination with nivolumab were also discontinued following reassessment. These unfavorable outcomes highlight the unresolved aspects of IDO1 function in cancer immune suppression [93].

Expression of IDO2 is observed in a more limited range of tissues than IDO1 (Table 2). In contrast to cervical cancer, overexpression of IDO2 is found in most cancers (e.g., GC, colon cancer, and renal tumors). IDO2 also inhibits T cell proliferation (in particular CD4+ and CD8+)  and, at the same time, increases the generation of Treg cells. Moreover, IDO2 supports cancer cell migration and proliferation, but its inhibition can reduce tumor volume and increase the number of tumor-infiltrating immune cells [94,95,96]. IDO2 might be a promising target for cancer immunotherapy, but more studies are needed to understand its role in cancer progression [70].

KMO is a key enzyme that controls the conversion of KYN to the important immunoregulatory and neuroactive TRP metabolites 3HKYN and QA. However, KMO inhibition leads to accumulation of KYN and increased KYNA level [83]. KMO is involved in inflammation and tumorigenesis. An imbalance in its activity could have clinical consequences and is associated with the pathogenesis of many types of cancer (Table 2).

KYNU is expressed in many organs (Table 2) and plays an essential role in inflammatory and cardiovascular diseases, as well as in various types of cancer, including glioma and osteosarcoma [97,98]. Moreover, KYNU is a promising therapeutic target in breast cancer, since its expression is positively associated with the expression of estrogen and progesterone receptors and E-cadherin and is negatively associated with the expression of HER2 [99]. KYNU expression is also higher in GC cells than in normal cells. Therefore, KYNU can promote the development of GC, and its overexpression is associated with a poor prognosis for patients. KYNU could be an interesting therapeutic target, as its deficiency can inhibit GC development [98].

Inhibition of ACMSD leads to increased brain levels of QA [81] and can play a crucial role in the development of brain tumors. For example, Guillemin et al. showed that, in a neuroblastoma cell line, the KP is disturbed. They observed weak expression of ACMSD in cancer cells, overproduction of QA (neurotoxin), and elevated consumption of PIC (antitumor and neuroprotective agent). The absence of ACMSD expression can contribute to tumor survival by direct TRP conversion to QA, subsequently leading to NAD+ production. Elevated NAD+ levels in cancer cells support their increased metabolic activity and promote their proliferation [100].

KAT is an enzyme that exists in humans, mice, and rats in four variants (KATI, II, III, and IV). There is interest in treating some brain disorders such as Alzheimer’s, dementia, and schizophrenia by targeting KAT. However, little information is available about the role of KAT in carcinogenesis. KAT is responsible for the production of KYNA, which promotes carcinogenesis and cancer progression (e.g., glioblastoma, colon adenocarcinoma, renal cell carcinoma, oral squamous cell carcinoma, colon adenocarcinoma, lung cancer). The administration of KAT inhibitors represents an interesting treatment strategy based on a decrease in endogenous KYNA production [101].

Although inhibition of 3-HAAO can reduce the amount of neurotoxic QA, it can result in the accumulation of 3HAA, which has proinflammatory properties [73]. Therefore, during inflammation, it plays a key role in the immune response. 3-HAAO overexpression leads to resistance of gliomas to chemoradiotherapy by protecting against oxidative stress and increasing NAD+ production. Patients with gliomas that overexpress 3-HAAO and QPRT have a poor prognosis [102]. QPRT catalyzes the transformation of QA into NAD+, the last step of the KP. NAD+ plays a key role in maintaining homeostasis in the human body, and dysregulation of its production leads to different states of disease, including cancers. Imbalanced expression of QPRT is also observed in several types of cancer (Table 2) and could be a prognostic factor for breast cancer and colon cancer [86]. The overexpression of QPRT correlates with a poor prognosis in patients with breast cancer [84,85,103]. QPRT activity enhances resistance to antileukemic drugs [104] and strengthens the resistance of malignant glioma [105], lung carcinoma [106], and colorectal cancer [107] cells to treatment. Glioblastoma cells were also found to use NAD+ (generated by QPRT activity) to repair the DNA damage caused by chemoradiotherapy [105].

## 6. Kynurenine Pathway Metabolites in Cancer Pathogenesis

### 6.1. Kynurenine

KYN plays a special role in this process, as it is an AHR ligand. AHR belongs to the basic helix–loop–helix–(bHLH) superfamily of transcription factors [19], which regulates biological processes responsible for maintaining tissue homeostasis or contributing to inflammation and cancer development [108]. Activation of KYN-AHR is correlated with enhanced levels of inflammatory cytokines (IL-1β, IL-6, IL-8) and a decrease in the number of infiltrating CD8+ cells around tumors. In the microenvironment, the persistence of elevated inflammatory cytokines causes chronic IDO1 expression in antigen-presenting cells (APC) such as macrophages, dendritic cells (DC), and/or tumor cells [15]. Overexpression of IDO1 leads to increased production of KP metabolites in the local cancer microenvironment, and consequently activation of AHR, which results in initiation of tumorigenesis by suppressing antitumor immune responses and promoting tumor cell survival and motility (Figure 4). However, immunosuppression caused by activation of the KYN-AHR pathway is accompanied by different mechanisms that arrest the production of effector T cells, promote IL10 production (potent anti-inflammatory cytokine), and enhance the proliferation of Treg cells, which support immune escape from cancer cells [109]. Notably, KYN is a precursor for other ligands for the AHR receptor (KYNA, XA, CA) [110]. AHR-KYN activation has been confirmed in human brain tumors, melanoma, renal cancer, B-cell lymphoma, Ewing sarcoma, bladder carcinoma, cervix carcinoma, colorectal carcinoma, lung carcinoma, and ovarian carcinoma, and is associated with tumor progression and drug resistance [111].

### 6.2. Kynurenic Acid

The potential role of KYNA in carcinogenesis is ambiguous. This metabolite interacts with different receptors (NMDA, alpha7 nAChR, GPR35, AHR) and several proteins involved in signal transmission and cell cycle regulation. Thus, KYNA can exert various biological effects on cancer cells and modulate the immune response to the initiation and progression of carcinogenesis [101]. GPR35 is highly expressed in immune cells (e.g., CD14+ monocytes, T cells, neutrophils, DCs) and expressed at lower levels in B-cells, eosinophils, basophils, platelets, and invariant natural killer T (iNKT) cells [112]. By activating the GPR35 receptor, KYNA reduces secretion of anti-inflammatory cytokines (IL-4) by NK cells [112]. In addition, through activation of GPR35, KYNA suppresses the production of IL-17 and IL-23 by CD4+ T cells and DCs [113]. Importantly, the neuroprotective and anti-inflammatory activity of KYNA has been attributed to its antagonism of NMDA receptors [114,115].

Significantly higher concentrations of this KP metabolite have been reported in cancer tissue or body fluids collected from patients with colon adenocarcinoma, oral squamous cell carcinoma [116], multiple myeloma [117], renal cell carcinoma (RCC) [118], myelodysplastic syndrome [119], primary cervical cancer [120], glioblastoma [121], and lung adenocarcinoma than in healthy tissue. Interestingly, KYNA has been proposed as a marker for non-small cell lung carcinoma, since the blood level of this metabolite in patients is significantly higher than that in healthy controls and correlates with disease progression [122]. An in vitro study showed a stimulatory effect of KYNA on the proliferation rate of mouse microglia and human glioblastoma cells [123]. On the other hand, there are also reports confirming an antiproliferative effect of KYNA against RCC cells [118] and demonstrating its inhibitory effect on the migration of glioblastoma cells [124].

### 6.3. 3-Hydroxyanthranilic Acid

Although 3HAA has been identified as a powerful antioxidant of reactive oxygen species [125], among KP metabolites it has the highest capacity to modulate the immune system. It shows anti-inflammatory activity by suppressing the secretion of T cell cytokines [126]. Furthermore, 3HAA induces apoptosis in monocyte and macrophage cell lines [127] and also thymocytes. This metabolite induces apoptosis in lymphocytes (Tc, Th1, and effector T cells) under in vivo and in vitro conditions through the release of cytochrome c or the activation of caspase-8. The death of Th1 and Th2 cells might lead to immune deviation under pathologic conditions [128]. 3HAA can inhibit the function of Th2 cells and at higher concentrations lead to their death by inhibiting transcription factor (NF-κB), which plays a critical role in the regulation of the survival, activation, and differentiation of innate immune cells and inflammatory T cells [129]. Furthermore, 3HAA may induce an increase in the number of Treg cells. This could impair the immune response to tumorigenesis. Furthermore, 3HAA significantly reduces secretion of the pro-inflammatory factors TNF, IL-6, IL-2, IL-5, IL-21, and MCP-1 by macrophages, proliferating macrophages, and DCs [126]. Thus, chronic generation of 3HAA by cancer cells could support tumor immune escape.

### 6.4. Quinolinic Acid

QA is involved in tumorigenesis and persistent cancer. Similar to 3HAA, QA modulates the immune response through selective inhibition of Th1 activation and proliferation [128]. QA expression can also increase the Treg population [130]. Notably, QA is a substrate for the synthesis of NAD+, which is rapidly consumed by cancer cells during tumorigenesis, conferring glioma resistance to oxidative stress [105]. Furthermore, QA increases the proliferation rate of some human glioblastoma (U343MG) [123], colon cancer (HT-116, HT-29) [131], and neuroblastoma (SK-N-SH) cell lines [100]. Moreover, QA increases the production of PIC by glial cell lines, leading to resistance to chemotherapeutics [100]. Importantly, QA has diagnostic potential as a biomarker, since elevated levels in high-grade gliomas are associated with poor prognosis [105].

### 6.5. Picolinic Acid

PIC shows antagonistic properties against the toxic effects of QA and affects the immune response by activating the pro-inflammatory functions of macrophages [132]. Moreover, PIC derivatives exhibit antitumor activity. This metabolite can affect tumor growth [133,134] and decrease cancer cell proliferation both in vitro and in vivo [100]. However, human trials are required to confirm these possible anticancer mechanisms of action. On the other hand, PIC at micromolar concentrations was found to inhibit T cell proliferation in vitro [132]. Furthermore, under in vitro conditions, PIC and IFN-γ can act synergistically, leading to transcriptional activation of the inducible isoform of the nitric oxide synthase (iNOS) gene in macrophages, stimulating the production of nitric oxide, and simultaneously enhancing macrophages’ cytotoxic properties [81].

## 7. Kynurenine Pathway Metabolites and Enzymes in Gastric Cancer

Overexpression of KP enzymes has been associated with mastitis and the poorest prognosis in patients with GC [135,136,137]. The available data demonstrate the role of IDO overexpression in and the contribution of the KP to the development of different gastrointestinal tumors. Notably, IDO overexpression occurs in up to 90% of human GCs [138] and can promote gastric tumor immune escape, as well as metastasis [92,135,137,139]. The activity of this enzyme causes the accumulation of immunosuppressive KP metabolites, which play a role in shaping aggressive phenotypes in GC [140]. Furthermore, IDO expression might serve as a biomarker for a poor prognosis in patients with GC. A prospective study of patients with gastric adenocarcinoma showed that IDO has potential as a prognostic biomarker for overall cancer patients after gastrectomy [141]. An in vitro study also showed that knockdown of IDO decreases the expression of some genes (LOXL2, COL6A1, COL6A2, and COL12A1) that were upregulated in GC tissues and inhibits cancer cell migration [139]. Lu et al. demonstrated high IDO1 expression across molecular subtypes in EBV and MSI tumors. In addition, IDO1 expression was correlated with increased tumor-infiltrating lymphocytes. These findings suggest that IDO1 inhibitors may have subtype-specific therapeutic potential, particularly in EBV and MSI GC [142]. Other results also indicate that IDO1 could be used as a prognostic biomarker for GC and to predict the efficacy of neoadjuvant chemotherapy [143]. Another study showed that IDO is not only associated with poor prognosis and immunotolerance through attenuation of Treg activation, but also that its expression is positively correlated with TGF-β expression, and that TGF-β expression is positively correlated with FoxP3 expression in patients with stage three GC [143]. Li et al. showed that IDO interacts with immunocompetent cells and humoral immune factors (DC2s, CD4+/CD8+ T cells, Foxp3+ Tregs), which compose the tumor microenvironment that supports GC immune escape and progression [144]. Moreover, IDO1 contributes to the inflammatory response that mediates the loss of parietal cells, leading to gastric metaplasia, which precedes the development of GC [145]. In particular, enhanced expression of IDO was detected in human gastric mucosa infected with *H. pylori*. In this case, IDO can modulate the Th1/Th1 and Th17 immune pathways, ultimately lowering gastric inflammation and favoring the persistence of *H. pylori*. Moreover, patients with GC infected with *H. pylori* exhibit a higher serum [KYN]/[TRP] ratio. This suggests that IDO activity supports tumor growth in patients with bacterial infection [135].

Previous research noted that IDO1 inhibitor monotherapies have low efficacy. However, combinations of IDO1 inhibitors with conventional treatments show very promising results. Currently, two phase two clinical trials are in progress. The first evaluates the benefit of epacadostat as an IDO1 inhibitor plus pembrolizumab in combination to treat patients with gastroesophageal junction cancer or GC, and the second evaluates the efficiency of BMS-986205 for the treatment of advanced GC [146].

In addition to IDO, other key KP enzymes such as KYNU or TDO2 could be interesting targets for therapy in GC. Shen et al. showed that KYNU knockdown significantly inhibited GC cell proliferation and invasion [136]. Its elevated expression was associated with aggressive phenotypes, which could lead to poor prognosis [136]. Liu et al. also reported that KYNU overexpression promotes GC cell proliferation, invasion, and metastasis [147]. In particular, elevated KYNU expression can lead to a decrease in KYN, a TRP metabolite essential for tumor development regulation. The results of a recent study showed that glutathione peroxidase 2 (GPx2) contributes to the progression and metastasis of GC by enhancing KYNU–KYN–AhR signaling pathway activity [111,148].

Similarly to IDO1, the expression of TDO2 in GC cells and tissue is significantly enhanced. A study by Pham et al. suggested that TDO2 could be a crucial marker to predict GC prognosis. Down-regulation of TDO2 expression inhibits GC cell proliferation, colony formation, and invasion. Furthermore, the expression of this enzyme is associated with cancer progression and overall clinical outcomes. Importantly, TDO2 expression correlates with tumor expression of PD-L1 and immune cell infiltration [149].

So far, few studies have reported the potential of KP metabolites to become GC biomarkers, but more evidence is still needed. In these studies, the concentrations of TRP and different KP metabolites were monitored in biofluids collected from GC patients, such as gastric juice, serum, plasma, and peritoneal fluid. However, other types of biofluids (such as salvia and urine) have not been screened (Figure 5). The main findings, which show the potential to develop a new diagnostic assay for GC detection and monitoring its progression [21,22,23,24,150], are summarized in Table 3 and Figure 5.

The high activity of IDO results in increased KYN levels in various cancers, including GC [141]. Wu and Wang showed that serum KYN levels are increased and associated with Treg cell proportions in chemotherapy-resistant GC patients. KYN secretion by GC cells may enhance chemoresistance by activating Tregs and the IL-10/STAT3/BCL2 signaling pathway in patients with GC [151]. According to a study by Cui et al., KYN, produced primarily by IDO1 in GC cells, reduced NK cell viability within the tumor microenvironment [152]. Recently, the KP was confirmed to be involved in the development of pre-cancerous changes, including gastric intestinal metaplasia. Liang et al. reported *H. pylori*–induced gastric intestinal metaplasia by KP activation in gastric epithelial cells. *H. pylori* was observed to promote TRP metabolism via KP and XA production. The alteration of metabolic reprogramming plays a crucial role in the development of *H. pylori*–related gastric disease. These data confirm that alteration of metabolic reprogramming plays a crucial role in the development of GC [153].

## 8. Conclusions

GC is currently one of the most common causes of cancer-related deaths, and generally has a poor prognosis. Early detection of GC is critical to increase patient survival rates and recovery chances. To achieve this goal, standardized recommendations for preventive screening and diagnostic tools that monitor therapeutic intervention must be established. Furthermore, no approved clinical marker allows accurate early diagnosis of this disease. In the case of GC, early prevention and diagnosis (e.g., blood biomarker testing) are likely to be much more effective than drug development. To better direct care, it is therefore crucial to identify new GC biomarkers, and especially to explore predictive (rather than prognostic) factors. Currently, researchers are focusing on searching for tumor markers in body fluids and developing a non-invasive technique characterized by repeated sampling, real-time monitoring, and precise treatment.

Recent advances in the development of analytical methods and omics analysis have led to the identification of new putative predictive and prognostic factors, but their clinical utility has to be verified. Investigation of imbalances in TRP metabolism via the KP could lead to new opportunities in the context of GC. Monitoring the expression of KP enzymes or the production of individual metabolites could be a promising GC diagnostic tool for use in precision oncology. Furthermore, targeted knockdown of the activity of KP enzymes could be an effective part of antitumor therapy. Simultaneous monitoring of KP constituents and other established or putative clinical markers (e.g., CEA, CA19-9 CA72-4, HER2) is also worth considering for antitumor therapy against gastric cancer. However, the topic remains largely unexplored. Future achievements in this field will aid in better understanding gastric cancer mechanisms or new personalized therapies.

## Figures and Tables

**Figure 1 ijms-26-03706-f001:**
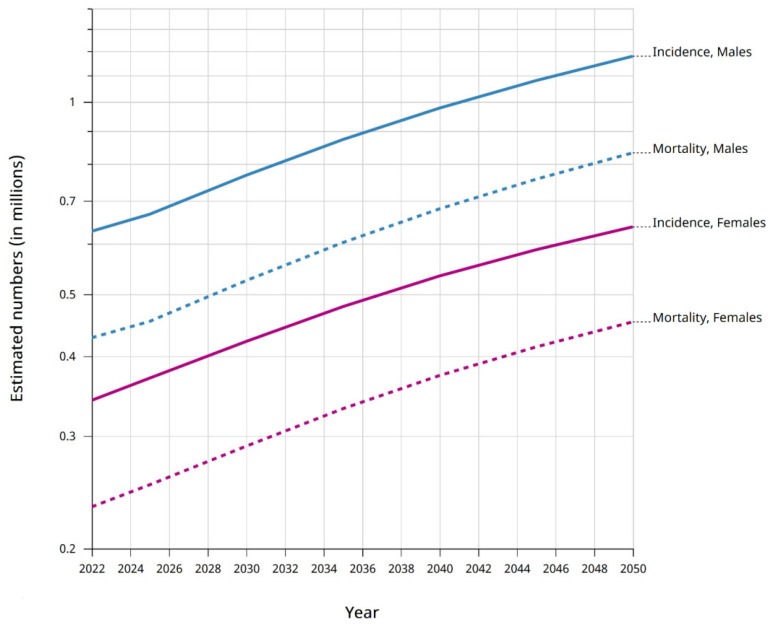
Estimated number of incidents and deaths related to GC in women and men aged 50–85+ years in 2022–2050 (graphic generated by the Cancer Tomorrow website [5], accessed February 2024).

**Figure 2 ijms-26-03706-f002:**
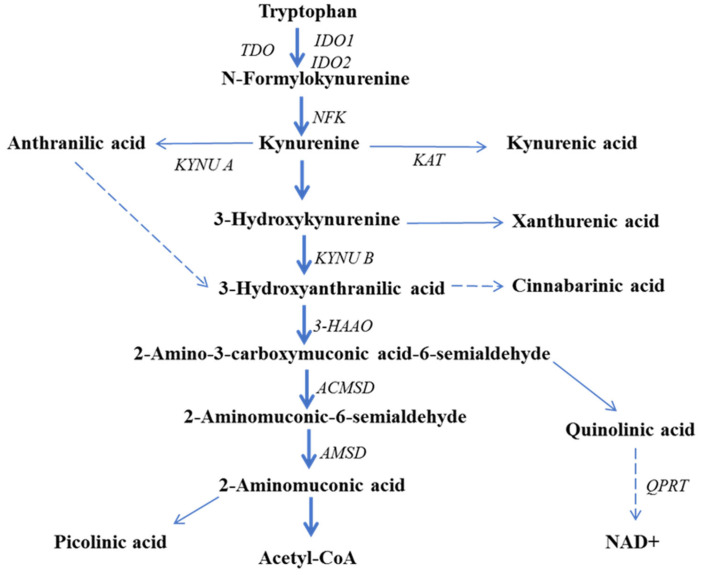
The kynurenine pathway of tryptophan degradation. Abbreviations: Acetyl-CoA—acetyl coenzyme A; ACMSD—2-amino-3-carboxymuconic acid semialdehyde decarboxylase; AMSD—2-aminomuconic semialdehyde dehydrogenase; IDO1/2—indoleamine-2,3-dioxygenase 1/2; KAT—kynurenine aminotransferase; KMO—kynurenine monooxygenase; KYNU A/B—kynureninase A/B; NAD—nicotinamide adenine dinucleotide; NFK—N′-formylkynurenine formamidase; TDO—tryptophan 2,3-dioxygenase; QPRT—quinolinate phosphoribosyl transferase; 3-HAAO—3-hydroxyanthranilic acid 3,4-dioxygenase.

**Figure 3 ijms-26-03706-f003:**
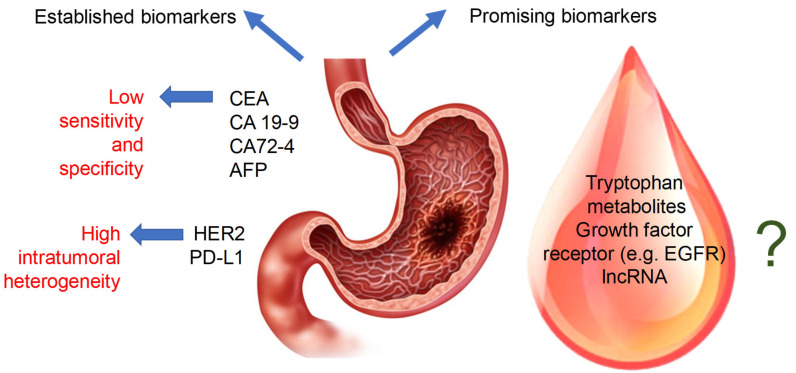
Established and promising blood biomarkers of gastric cancer.

**Figure 4 ijms-26-03706-f004:**
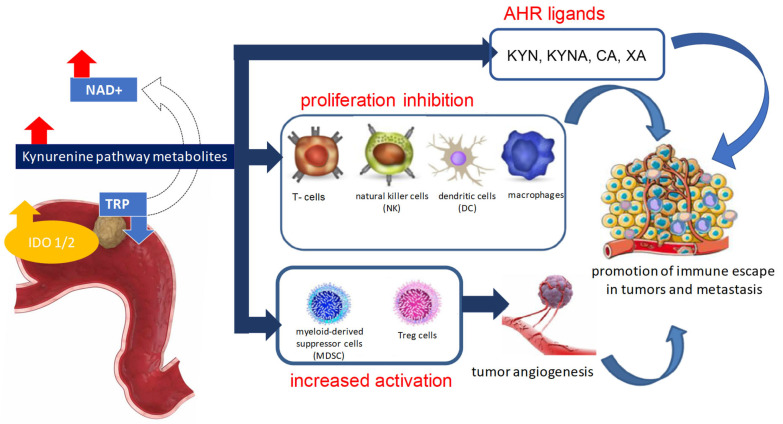
Scheme presenting the involvement of tryptophan metabolism via the kynurenine pathway in gastric cancer progression.

**Figure 5 ijms-26-03706-f005:**
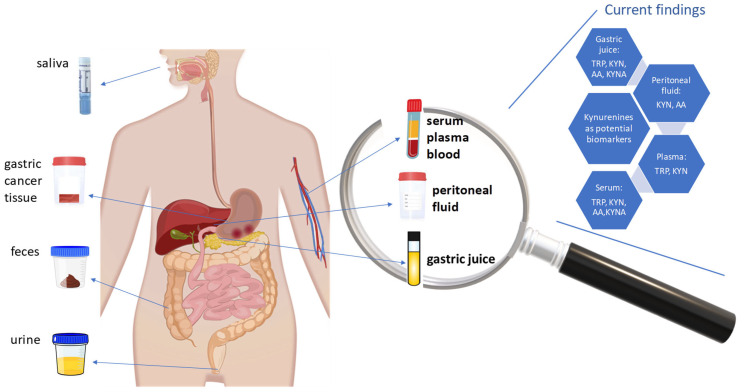
Types of biological material that could be used to evaluate the role of kynurenine pathway metabolites as gastric cancer biomarkers.

**Table 1 ijms-26-03706-t001:** Examples of molecular biomarkers of gastric cancer with potential clinical value.

Biomarker	Type	Role in Carcinogenesis	Potential Clinical Value	Material for Analysis	Detection Method	Ref.
B7–H3	Transmembrane protein	It is involved in cancer progression and metastasis and may inhibit CD8+ T cells	Predictive biomarker	Gastric tissue	IHC	[30]
CLDN 18.2	Claudin family protein	It is involved in cancer cell proliferation, invasion, and metastasis	Prognostic and therapeutic biomarker	Gastric tissue	IHC	[46]
E-Cadherin	Transmembrane glycoprotein	It is involved in cancer cell proliferation, invasion, and metastasis. Reduces response to conventional and targeted therapy	Prognostic and predictive biomarker	Gastric tissue	IHC	[47,48]
EGFR	Growth factor receptor	Regulates cell proliferation, migration, survival, and tumor angiogenesis. Amplified or overexpressed in intestinal-type GC	Prognostic biomarker	Gastric tissue, serum exosomes	IHC	[9,30,49]
FGFR2	Growth factor receptor	Its overexpression is associated with greater depth of tumor invasion, higher rates of lymph node metastasis, more advanced disease stage, and worse outcome. It is amplified or overexpressed in diffuse-type GC	Predictive biomarker	Tissue from distant metastases, primary tumors	IHC, NGS, ISH	[9,30,46,50]
GST-π	Enzyme	It is associated with tumor invasion and recurrence	Predictive biomarker	Gastric tissue	IHC	[51,52]
lncRNA	RNA molecule	Its expression regulates GC cell proliferation, cell cycle, apoptosis, invasion, migration, metastasis, and tumorigenicity	Diagnostic, predictive, and therapeutic biomarker. Promising biomarker for early diagnosis	Serum exosomes, plasma	qRT-PCR	[53,54]
MET	Hepatocyte growth factor receptor	Induces proliferation, survival, motility, cell scattering, angiogenesis, and tubulogenesis. Drives epithelial–mesenchymal transition and tumor invasion. Amplified or overexpressed in diffuse-type GC	Predictive and therapeutic biomarker	Gastric tissue	IHC, FISH, NGS	[9,30,46,55]
MRP2	Protein	Involved in mechanisms mediating multidrug resistance	Predictive biomarker	Gastric tissue	IHC	[52]
mTOR	Kinase	Involved in cell growth, differentiation, and survival. Amplified or overexpressed in intestinal-type and diffuse-type GC	Therapeutic target	Gastric tissue	IHC	[56]
MUC1	High molecular weight glycoprotein	Oncoprotein involved in tumor proliferation, metabolism, invasion, and metastasis	Prognostic biomarker	Gastric tissue	IHC	[57,58]
SOX9	Transcription factor	Critical for GC cell survival and promotes cancer cell proliferation and chemoresistance. Its expression is related to tumor progression and associated with advanced stages, lymph node metastasis, and extra-capsular growth in lymph node metastasis	Prognostic biomarker	Gastric tissue	IHC	[9,59]
T53	Nuclear protein	Supports cancer progression. Tumor suppressor	Prognostic and predictive biomarker	Gastric tissue	IHC	[48,60]
VEFGR	Growth factor receptor	Involved in angiogenesis and lymphangiogenesis	Prognostic, therapeutic, and predictive biomarker	Gastric tissue, serum	IHC	[46]
VISTA	Membrane protein	Suppresses T cell activation. Supports immune escape	Predictive biomarker	Gastric tissue	IHC	[30]

EGFR—epidermal growth factor receptor; FGFR2—fibroblast growth factor receptor 2; FISH—fluorescence in situ hybridization; GST-π—glutathione S-transferase; ISH—in situ hybridization; IHC—immunohistochemistry; lncRNA—long non-coding ribonucleic acid; MET—proto-oncogene c-mesenchymal-epithelial transition; MRP2—multidrug resistance-associated protein 2; mTOR—mammalian target of rapamycin; MUC1—transmembrane glycoprotein Mucin 1; qPCR—quantitative real-time reverse transcription-polymerase chain reaction; q-RT-PCR—real-time quantitative reverse transcription polymerase chain reaction; VEFGR—vascular endothelial growth factor; VISTA—V-domain immunoglobulin-containing suppressor of T cell activation.

**Table 2 ijms-26-03706-t002:** Enzymes in the kynurenine pathway: localization, main inhibitors, and activators.

Enzyme	Localization	Activators	Inhibitors	Up-Regulation or Down-Regulation in Disorders	Ref.
IDO1	Placenta, colon, epididymis, dendritic cells, macrophages, reticular cells, cancer cells, innate immune cells, stromal cells, brain, kidney, lung, spleen	IFN-γ, IFN-α, TNF-α, IL-1, IL-12, IL-18, IL-6, IL-10	Nitric oxide, TRP excess (>50 µM)	Cervical carcinomas, endometrial carcinomas, bladder carcinomas, kidney carcinomas, non-small cell lung carcinomas, ovarian carcinomas, melanomas, stomach carcinomas, colorectal carcinomas, head and neck carcinomas, esophageal carcinomas, prostate carcinomas, breast carcinomas, pancreatic carcinomas, glioblastomas	[18,64,65,66,67,68,69]
IDO2	Brain, liver, kidney, epididymis, dendritic cells, B-cells, placenta, epididymis	IFN-γ, IL-10, PGE2, lipopolysaccharide	1-methyl-DL-tryptophan	Non-small cell lung cancer, pancreatic cancer, colon cancer, gastric cancer, renal tumors	[64,65,70,71,72]
TDO	Liver, brain kidney, skin, placenta, pregnant uterus, epididymis, testis	Glucocorticoids, TRP substrates, estrogens, heme cofactor	NAD(P)H-mediated feedback mechanism, progesterone, estrogens, IFN-γ	Bladder carcinoma, brain tumor, breast carcinoma, cervix carcinoma, colorectal carcinoma, Ewing sarcoma, head and neck carcinoma, hepatocarcinoma, leukemia, lung, carcinoma, melanoma, mesothelioma, B-cell lymphoma, neuroblastoma, ovarian carcinoma, renal cell carcinoma, sarcoma	[14,19,51,62,73,74]
NFK	Liver, kidney, brain	*o*-aminophenol, *o*-aminotoluidine	Organophosphate, insecticides, metal cations	Colon cancer	[16,73,75,76]
KMO	The outer membrane of mitochondria, liver, kidney, macrophages and monocytes, central nervous system, placenta	Inflammatory cytokines, IFN-γ, ROS	4-aryl-4-oxobutanoic acids, sulfonamides, 6-phenylpyrimidines, phenyloxadiazoles, riboflavin, excess of TRP, AA, XA	Schizophrenia, infectious diseases, renal clear cell carcinoma, lower-grade brain glioma, acute myeloid leukemia, TNBC, hepatocellular carcinoma	[16,63,73,75,77,78,79,80]
KYNU	Liver, kidney	IFN-γ	Vitamin B6 deficiency, leucine-rich diets	Chronic inflammatory skin diseases, many autoimmune and autoinflammatory diseases, renal papillary cell carcinoma, ovarian serous cyst adenocarcinoma, pancreatic adenocarcinoma, lung adenocarcinoma, lower-grade brain glioma, renal papillary cell carcinoma, acute myeloid leukemia	[14,16,63,73]
KAT	Kidney, placenta, heart, macrophages, liver, brain	KYN, 2-oxoglutarate, pyruvate, oxaloacetic acid	Vitamin B6 deficiency, IFN-γ	Schizophrenia	[16,73,74,77]
3HAAO	Liver, kidney, brain	IFN-γ, Fe^2+^	Iron chelators	Neurological disorders, esophageal carcinoma	[16,63,72]
ACMSD	Kidney, liver	Leucine, Fe^2+^, Co^2+^	QA, PIC, KYNA, glycolytic intermediates: glyceraldehyde-3-phosphate, 3-phosphoglycerate, phosphoenolpyruvate, 2-phosphoglycerate, some metal ions: Zn^2+^, Fe^3+^, Cr^3+^, Cd^2+^	Brain tumors, neuroinflammatory diseases	[16,75,81,82]
QPRT	Liver, kidney, brain	-	Phthalic acid, O_2_, pyridine analogs of QA, lysine, histidine, arginine, some cations (Cu^2+^, Fe^2+^, Fe^3+^), various carboxylic acids, IFN-γ	Inflammatory diseases, glioma, breast cancer, stomach adenocarcinoma, cutaneous melanoma, glioblastoma, colon cancer	[16,73,75,83,84,85,86]

AA—anthranilic acid; ACMSD—2-amino-3-carboxymuconate-6-semialdehyde decarboxylase, AhR—aryl hydrocarbon receptor; ATP—adenozyno-5′-trifosforan; IDO—indoleamine 2,3-dioxygenase; IL—interleukin; IFN—interferon; KAT—kynurenine aminotransferase; KMO—kynurenine monooxygenase; KYN—kynurenine; KYNA—kynurenic acid; NADPH—nicotinamide adenine dinucleotide phosphate hydrogen; NFK—N′-formylkynurenine formamidase; PGE2—prostaglandin E2; PIC—picolinic acid; TDO—tryptophan 2,3-dioxygenase; TNBC—triple-negative breast cancer; TNF—tumor necrosis factor; TRP—tryptophan; QA—quinolinic acid; QPRT—quinolinate phosphoribosyl transferase; ROS—reactive oxygen species; XA—xanthuerenic acid; 3HAAO—3-hydroxyanthranilic acid oxygenase.

**Table 3 ijms-26-03706-t003:** Current achievements in the study of the role of kynurenine pathway metabolites in gastric cancer.

Metabolite	Sample Type	Major Findings	Ref.
TRP, KYN	Plasma	Statistically significant lower relative level of TRP in samples from patients with GC compared to *H. pylori*–negative patients with NAG-, *H. pylori*–positive patients with CAG+, and patients having precursor lesions of GC (atrophy and/or intestinal metaplasia, PLGC). Slightly higher relative concentration of KYN in plasma from patients with GC than in NAG-, CAG+, and PLGC samples (not statistically significant differences).	[150]
TRP, KYN	Serum	Significant decrease in TRP level and not statistically significant increase in KYN level in serum from patients with GC than in controls without malignant disease. Significant increase in [KYN]/[TRP] ratio in serum from GC patients. In the *H. pylori*–seronegative subgroup, there were no significant differences in serum KYN and [KYN]/[TRP] ratio, and a significant decrease in serum TRP levels, between cancer-free individuals and patients with GC. In the *H. pylori*–seropositive subgroup, there was a significant increase in serum [KYN]/[TRP] ratio and a decrease in TRP serum levels in patients with GC compared to controls. Significant correlation between serum neopterin levels and [KYN]/[TRP] ratio in *H. pylori*–seronegative and –seropositive controls and GC individuals.	[24]
TRP, KYN, AA, KYNA	Serum, gastric juice	Statistically significant increase in AA and KYNA levels, and a significant decrease in KYN levels, in serum from the GC group compared with the controls. Comparable serum TRP levels between the studied groups. Statistically significant increase in TRP, AA, and KYNA levels in gastric juice from patients with GC compared with the control group. The KYN level also increases, but with no statistical significance.	[21]
KYN, AA	Serum, peritoneal fluid, lavage washings	Significantly higher AA levels in peritoneal lavage washings in patients with pN1-3 compared to those with pN0. Positive correlations between AA level in peritoneal fluid with pN stage and between AA level in peritoneal lavage washings with cT stage. Significantly higher KYN levels in peritoneal lavage washings in cM1 patients than in cM0 patients. Positive correlation between KYN level with cM (peritoneum) stage. Positive correlation of KYN level with cT and negative correlation of 3HKyn and XA levels with cM in patients with GC.	[23]

AA—anthranilic acid; CAG+—chronic active gastritis; GC—gastric cancer; KYN—kynurenine; KYNA—kynurenic acid; NAG-—non-active gastritis; TRP—tryptophan; XA—xanthurenic acid; 3HKYN—3-hydroxykynurenine.

## Data Availability

No data were used for the research described in the article.

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
