# Peer review of "Promising Gastric Cancer Biomarkers—Focus on Tryptophan Metabolism via the Kynurenine Pathway"

_ijms, 2025, doi:10.3390/ijms26083706_

Round 1

Reviewer 1 Report

Comments and Suggestions for Authors

His review discusses predictive, prognostic, and diagnostic markers for gastric cancer, focusing on the role of tryptophan metabolites from the kynurenine pathway as prognostic and predictive factors. The text is well-structured, and there are no significant issues; however, some concerns are noted:

  1. There is insufficient description of early gastric cancer diagnosis through endoscopic methods. It would be beneficial to include evidence on early gastric cancer diagnosis using the Kyoto classification of gastritis and IEE (NBI or BLI). Additionally, mentioning recent advancements in diagnostic approaches using capsule endoscopy for the stomach would enhance the content.
  2. The authors point out that understanding the tryptophan metabolism pathway could lead to new therapeutic strategies for various cancers, including gastric cancer. While developing therapies targeting the regulation of the kynurenine pathway or inhibition of IDO is promising, further clinical considerations on this topic would be valuable.

Comments on the Quality of English Language

Minor English editing is required, 

Author Response

Thank you very much for your meticulous reading of our manuscript and for providing all suggestions to improve the presented paper. We have addressed all issues and provided our answers below. We hope to have addressed correctly your comments.

Substantial changes were made and they are highlighted in yellow. Furthermore, we have read the manuscript thoroughly and corrected typing and grammatical errors were using an institutional instance of Writefull Revise software. We hope that the new version of the manuscript will be satisfactory, and now meets the requirements for publishing in International Journal of Molecular Sciences.

Responses to Reviewers’ Comments:

Reviewer #1:

His review discusses predictive, prognostic, and diagnostic markers for gastric cancer, focusing on the role of tryptophan metabolites from the kynurenine pathway as prognostic and predictive factors. The text is well-structured, and there are no significant issues; however, some concerns are noted:

Response: Thank you for appreciate our work.

There is insufficient description of early gastric cancer diagnosis through endoscopic methods. It would be beneficial to include evidence on early gastric cancer diagnosis using the Kyoto classification of gastritis and IEE (NBI or BLI). Additionally, mentioning recent advancements in diagnostic approaches using capsule endoscopy for the stomach would enhance the content.

Response: Thank you for your valuable comment. We addressed this concern by supplementing the information on endoscopy in accordance with the Reviewer's suggestion. New text was added in the section 2: Gastric cancer etiology and diagnosis. We also added new references:

Toyoshima O, Nishizawa T, Koike K. Endoscopic Kyoto classification of Helicobacter pylori infection and gastric cancer risk diagnosis. World J Gastroenterol. 2020, 26(5), 466–477.

Dohi O, Seya M, Iwai N, Ochiai T, Yumoto J, Mukai H, et al. Endoscopic detection and diagnosis of gastric cancer using image‐enhanced endoscopy: A systematic review and meta‐analysis. DEN Open. 2025, 1-10.

Huang Y, Shao Y, Yu X, Chen C, Guo J, Ye G. Global progress and future prospects of early gastric cancer screening. J Cancer. 2024, 15(10), 3045–3064.

The authors point out that understanding the tryptophan metabolism pathway could lead to new therapeutic strategies for various cancers, including gastric cancer. While developing therapies targeting the regulation of the kynurenine pathway or inhibition of IDO is promising, further clinical considerations on this topic would be valuable.

Response: Thank you for your valuable comment. We added information about clinical trial on IDO inhibitors in section 5. ‘The role of the kynurenine pathway enzymes in tumorigenesis’. Furthermore, we addressed this issue in section 7. ‘Kynurenine pathway metabolites and enzymes in gastric cancer’. The discussion on this topic was supported by the following references:

Liu M, Wang X, Wang L, Ma X, Gong Z, Zhang S, et al. Targeting the IDO1 pathway in cancer: from bench to bedside. J Hematol OncolJ Hematol Oncol. 2018, 11(1), 1-12.

Yan J, Chen D, Ye Z, Zhu X, Li X, Jiao H, et al. Molecular mechanisms and therapeutic significance of Tryptophan Metabolism and signaling in cancer. Mol Cancer. 2024, 23(1), 1-32.

Xiang Z, Li J, Song S, Wang J, Cai W, Hu W, et al. A positive feedback between IDO1 metabolite and COL12A1 via MAPK pathway to promote gastric cancer metastasis. J Exp Clin Cancer Res. 2019, 38(1), 1-12.

Liu H, Shen Z, Wang Z, Wang X, Zhang H, Qin J, et al. Increased expression of IDO associates with poor postoperative clinical outcome of patients with gastric adenocarcinoma. Sci Rep. 2016, 6(1), 1-9.

Larussa T, Leone I, Suraci E, Nazionale I, Procopio T, Conforti F, et al. Enhanced Expression of Indoleamine 2,3-Dioxygenase in Helicobacter pylori-Infected Human Gastric Mucosa Modulates Th1/Th2 Pathway and Interleukin 17 Production. Helicobacter. 2015, 20(1), 41–48.

Liu, X. H., Zhai, X. Y. Role of tryptophan metabolism in cancers and therapeutic implications. Biochimie. 2021, 182, 131-139.

Shen K, Chen B, Yang L, Gao W. KYNU as a Biomarker of Tumor-Associated Macrophages and Correlates with Immunosuppressive Microenvironment and Poor Prognosis in Gastric Cancer. Int J Genomics. 2023, 2023, 1-31.

Pham QT, Taniyama D, Akabane S, Takashima T, Maruyama R, Sekino Y, et al. Essential Roles of TDO2 in Gastric Cancer: TDO2 Is Associated with Cancer Progression, Patient Survival, PD-L1 Expression, and Cancer Stem Cells. Pathobiology. 2023, 90(1), 44–55.

Mansorunov D, Apanovich N, Kipkeeva F, Nikulin M, Malikhova O, Stilidi I, et al. The Correlation of Ten Immune Checkpoint Gene Expressions and Their Association with Gastric Cancer Development. Int J Mol Sci. 2022, 23(22), 1-14.

Lu S, Wang LJ, Lombardo K, Kwak Y, Kim WH, Resnick MB. Expression of Indoleamine 2, 3-dioxygenase 1 (IDO1) and Tryptophanyl-tRNA Synthetase (WARS) in Gastric Cancer Molecular Subtypes. Appl Immunohistochem Mol Morphol. 2020, 28(5), 360-368.

Nishi M, Yoshikawa K, Higashijima J, Tokunaga T, Kashihara H, Takasu C, et al. The Impact of Indoleamine 2,3-dioxygenase (IDO) Expression on Stage III Gastric Cancer. Anticancer Res. 2018, ;38(6), 3387–3392.

El-Zaatari M, Bass AJ, Bowlby R, Zhang M, Syu LJ, Yang Y, et al. Indoleamine 2,3-Dioxygenase 1, Increased in Human Gastric Pre-Neoplasia, Promotes Inflammation and Metaplasia in Mice and Is Associated With Type II Hypersensitivity/Autoimmunity. Gastroenterology. 2018, 154(1),140-153.

Reviewer 2 Report

Comments and Suggestions for Authors

The introduction does a very good job in bringing attention to the importance of diagnostic and prognostic factors in advanced stage gastric cancer.. Gastric cancer more frequently diagnosed in advanced stages, and as the authors stated, further research is imperative in treating these malignancies. The information is presented clearly throughout the article, with use of many tables and images to increase understanding of the detailed subject matter. The biomarkers are presented and discussed in a manner that makes for an easy understanding for the reader and the English language used is correct in terms of word choice and grammar. Overall. the article is well written and provides an interesting take of the diagnosis and prognosis of advanced stage gastric cancer in terms of biomarkers. 

Author Response

Thank you very much for your meticulous reading of our manuscript and for providing all suggestions to improve the presented paper.  Substantial changes were made and they are highlighted in yellow. Furthermore, we have read the manuscript thoroughly and corrected typing and grammatical errors were using an institutional instance of Writefull Revise software. We hope that the new version of the manuscript will be satisfactory, and now meets the requirements for publishing in International Journal of Molecular Sciences.

Reviewer #2:

The introduction does a very good job in bringing attention to the importance of diagnostic and prognostic factors in advanced stage gastric cancer. Gastric cancer more frequently diagnosed in advanced stages, and as the authors stated, further research is imperative in treating these malignancies. The information is presented clearly throughout the article, with use of many tables and images to increase understanding of the detailed subject matter. The biomarkers are presented and discussed in a manner that makes for an easy understanding for the reader and the English language used is correct in terms of word choice and grammar. Overall. the article is well written and provides an interesting take of the diagnosis and prognosis of advanced stage gastric cancer in terms of biomarkers. 

Response: Thank you for appreciate our work.

Reviewer 3 Report

Comments and Suggestions for Authors The review titled " Promising biomarkers of gastric cancer – focused on tryptophan metabolism via the kynurenine pathway" focuses its attention on the role of tryptophan of tryptophan metabolites from the kynurenine pathway as prognostic, predictive, and diagnostic factors in gastric cancer . The paper is well written with a good structure and is clear in its exposition even for young readers.   A careful study of the review as a whole is very interesting but its objective should be changed, especially considering that out of 7 paragraphs only one, and in a marginal way, takes into consideration the role of tryptophane in gastric cancer. If the evidence, as shown, is limited, the review should concern the aspect of typtophane metabolism in cancer in a general way and not add the role in gastric cancer as aim deduced from the title. Although there is a broad introduction in gastric cancer and all its implications, the evidence of tryptophane metabolism is still too preliminary to give it due weight, it would be interesting instead to investigate the role of tryptophane metabolism in cancer in general.   I suggest deleting or merging paragraphs 6.3 , 6.4.3, 6.5, 6.6, they are not very influential on the objective of the paper.   Improve the indexing of paragraphs.   Check typing and grammatical errors.     Comments on the Quality of English Language

None

Author Response

Thank you very much for your meticulous reading of our manuscript and for providing all suggestions to improve the presented paper. We have addressed all issues and provided our answers below. We hope to have addressed correctly your comments.

Substantial changes were made and they are highlighted in yellow. Furthermore, we have read the manuscript thoroughly and corrected typing and grammatical errors were using an institutional instance of Writefull Revise software. We hope that the new version of the manuscript will be satisfactory, and now meets the requirements for publishing in International Journal of Molecular Sciences.

Reviewer #3:

The review titled " Promising biomarkers of gastric cancer – focused on tryptophan metabolism via the kynurenine pathway" focuses its attention on the role of tryptophan of tryptophan metabolites from the kynurenine pathway as prognostic, predictive, and diagnostic factors in gastric cancer . The paper is well written with a good structure and is clear in its exposition even for young readers.

Response: Thank you for appreciate our work.

A careful study of the review as a whole is very interesting but its objective should be changed, especially considering that out of 7 paragraphs only one, and in a marginal way, takes into consideration the role of tryptophane in gastric cancer. If the evidence, as shown, is limited, the review should concern the aspect of typtophane metabolism in cancer in a general way and not add the role in gastric cancer as aim deduced from the title. Although there is a broad introduction in gastric cancer and all its implications, the evidence of tryptophane metabolism is still too preliminary to give it due weight, it would be interesting instead to investigate the role of tryptophane metabolism in cancer in general.

Response: Thank you for your valuable comment. We significantly extended the paragraph 7 on new information to underline the promising importance of targeting kynurenine pathway metabolites and enzymes in gastric cancer. In addition to KP metabolites, we decided to discuss the role of KP enzymes in gastric cancer to provide more comprehensive overlook. As a result of the changes we have made, we have decided to slightly change the title of paragraph 7 to: ‘Kynurenine pathway metabolites and enzymes in gastric cancer’. We also added information about the clinical trials on IDO inhibitors in terms of gastric cancer treatment. The discussion contained in this paragraph was conducted on the basis of 24 references. We hope that the current version of this paragraph better summarizes the state of knowledge on this topic.

I suggest deleting or merging paragraphs 6.3 , 6.4.3, 6.5, 6.6, they are not very influential on the objective of the paper.

Response: Thank you for your valuable comment. We merged them into a single paragraph.

Improve the indexing of paragraphs.

Response: Thank you. Corrected.

Check typing and grammatical errors.

Response: Thank you, We have read the manuscript thoroughly and corrected typing. Grammatical errors were corrected using an institutional instance of Writefull Revise.

Round 2

Reviewer 3 Report

Comments and Suggestions for Authors -- I suggest to Delete the explanation of the technologies in the GC on pages 6 and 7 -- I suggest not to introduce miRNAs as potential biomarkers since a few words later it is stated that they are not very reliable and repeatable for clinical application -- I suggest to Delete paragraph 6.3 and I would consider keeping paragraphs 6.4/6.5/6.6

Author Response

Thank you very much for providing additional suggestions to improve our manuscript. We have addressed all the issues and provide our answers below. We hope to have addressed your comments correctly. Substantial changes were made, and they are highlighted in yellow and in track changes mode.

Responses to Reviewers’ Comments:

Reviewer #1:

- I suggest to Delete the explanation of the technologies in the GC on pages 6 and 7

Response: Thank you for your suggestion. We included information on different endoscopy techniques following the suggestion of another reviewer. In the current version, we have made corrections and shortened the descriptions of the endoscopic techniques used in GC diagnostics. We hope that these improvements are satisfactory.

- I suggest not to introduce miRNAs as potential biomarkers since a few words later it is stated that they are not very reliable and repeatable for clinical application

Response: Thank you for your valuable comment. We removed information on miRNA from the manuscript.

- I suggest to Delete paragraph 6.3 and I would consider keeping paragraphs 6.4/6.5/6.6

Response: Thank you for your suggestion. We removed paragraph 6.3. We would like to keep paragraphs 6.4, 6.5 and 6.6.

Round 3

Reviewer 3 Report

Comments and Suggestions for Authors

X